# DEEP EVIDENTIAL REINFORCEMENT LEARNING FOR DYNAMIC RECOMMENDATIONS

## ABSTRACT

Reinforcement learning (RL) has been applied to build recommender systems (RS) to capture users' evolving preferences and continuously improve the quality of recommendations. In this paper, we propose a novel deep evidential reinforcement learning (DERL) framework that learns a more effective recommendation policy by integrating both the expected reward and evidence-based uncertainty. In particular, DERL conducts evidence-aware exploration to locate items that a user will most likely take interest in the future. Two central components of DERL include a customized recurrent neural network (RNN) and an evidential-actor-critic (EAC) module. The former module is responsible for generating the current state of the environment by aggregating historical information and a sliding window that contains the current user interactions as well as newly recommended items that may encode future interest. The latter module performs evidence-based exploration by maximizing a uniquely designed evidential Q-value to derive a policy giving preference to items with good predicted ratings while remaining largely unknown to the system (due to lack of evidence). These two components are jointly trained by supervised learning and reinforcement learning. Experiments on multiple real-world dynamic datasets demonstrate the state-of-the-art performance of DERL and its capability to capture long-term user interests.

## 1 INTRODUCTION

Recommender systems (RS) have been widely used for providing personalized recommendations in diverse fields such as media, entertainment, and e-commerce by effectively improving user experience (Su & Khoshgoftaar, 2009; Sun et al., 2014; Xie et al., 2018). Various methods have been introduced to tackle the recommendation problem. Traditional methods include: *collaborative filtering*, which captures user preferences using information of similar users (Koren, 2008), *content-based*, where extra information is used for better latent preference and item representation (Mooney & Roy, 2000), and *hybrid*, which integrates both collaborative and content-based methods for a more effective recommendation (Burke, 2002). Deep learning (DL) has also been increasingly used to build RS due to its ability to model complex and non-linear user-item relationships (Cheng et al., 2016; Guo et al., 2017).

Most RS methods mentioned above consider recommendation as a static process, which fails to consider users' evolving preferences. Some efforts have been devoted to capture users' evolving preferences by shifting the user latent preference over time (Koren, 2009; Charlin et al., 2015; Gultekin & Paisley, 2014). Similarly, sequential recommendation methods (Kang & McAuley, 2018; Tang & Wang, 2018) attempt to incorporate users' dynamic behavior by leveraging previously interacted items. However, both static and dynamic recommendation methods primarily focus on maximizing the immediate (short-term) reward when making recommendations. As a result, they fail to take into account whether these recommended items will lead to long-term returns in the future, which is essential to maintain a stable user base for the system in the long run.

Several recent works have adapted reinforcement learning (RL) in the RS (Chen et al., 2019b; Zhao et al., 2017). RL has already gained huge success in diverse fields, such as robotics (Kober et al., 2013) and games (Silver et al., 2017). The core idea of RL is to learn an optimal policy to maximize the total expected reward in the long run. RL methods consider a recommendation procedure as sequential interactions between users and RL agents to learn the optimal recommendation policies effectively. Although RL approaches show promising results in RS (Chen et al., 2019b; Zheng et al.,

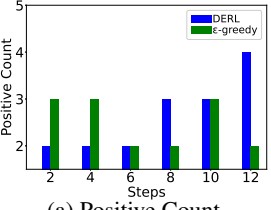 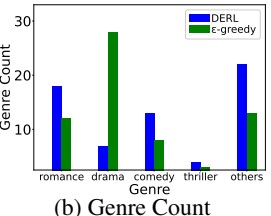 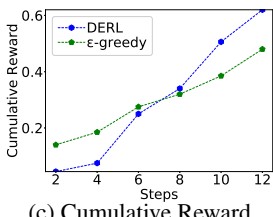

| (a) Positive Count | (b) Genre Count | (c) Cumulative Reward |
|---|---|---|

Figure 1: Different recommendation behavior between an existing RL model and DERL

2018), they primarily rely on standard exploration strategies ($\epsilon$-greedy), which are less effective in a large item space with sparse reward signals given the limited interactions for most users. Therefore, they may not learn the optimal policy that provides the most informative recommendations to capture effective user preferences and achieve maximum expected reward in the long run.

Figure 1 further illustrates the limitation of existing RL methods using a standard $\epsilon$-greedy strategy for exploration. The RL agent primarily focuses on highly-rated items in early steps as shown in Figure 1a. Most of these items come from the same genre as shown in Figure 1b, which is further verified by the detailed recommendation list given by Table 1. Such a recommendation behavior leads to a lower cumulative reward in the later steps as shown in Figure 1c. As Table 1 shows, $\epsilon$-greedy mostly

Table 1: Examples of recommended movies

| Model | Movies | Movie Genre |
|---|---|---|
| DERL | Sound of Music (1965) | **Musical** |
| | Casino (1995) | Drama |
| | Ben-Hur (1959) | **Action,Adventure** |
| | The Bug's Life (1998) | **Animation,Comedy** |
| | Babe (1995) | **Children's,Comedy** |
| $\epsilon$-greedy | Pocahontas (1995) | **Musical** |
| | Wizard of Oz (1939) | Drama |
| | Christmas Story (1983) | Drama |
| | Erin Brockovich (2000) | Drama |
| | Restoration (1995) | Drama |

focuses on Drama movies based on the user's current preference. It only captures one novel genre (i.e., Musical, bold in the table) that matches the user's long-term interest. It is clear that more systematic exploration is essential to discover users' long-term interests to maximize the future reward.

To address the above key challenges, we propose a novel deep evidential reinforcement learning (DERL) method that utilizes a balanced exploitation (with high predicted ratings) and exploration (with evidence-based uncertainty) strategy for effective recommendations. We formulate an evidential RL framework that augments the maximum reward RL objective with evidence-based uncertainty maximization. More importantly, the evidence-based uncertainty formulation substantially improves exploration and robustness by acquiring diverse behaviors that are indicative of a user's long-term interest. As shown in Figure 1b, DERL devotes a strong focus on more diverse genres (denoted by 'others' in the figure) and many of these capture the long-term interest from the user as verified by the detailed recommendation list in Table 1. In this case, we refer to a user's long-term interest as the genres of movies frequently watched by the user in the later phase of interactions (i.e., after time step 8 in the given example). These genres are Musical, Action, Adventure, Comedy, and Animation, which match almost perfectly with what DERL recommends in Table 1. DERL seamlessly integrates two major components: a customized *RNN* and an *Evidence-Actor-Critic* module. The former primarily focuses on generating the current state of the environment by aggregating the previous state, current items captured by a sliding window, and future recommended items from the RL agent. This provides effective means of dynamic state representation for better future recommendations. Meanwhile, the EAC module leverages evidence-based uncertainty to most effectively explore the item space to identify items that potentially align with the user's long-term interest. It encourages learning the optimal policy by maximizing a novel evidential Q-value to achieve a maximum long-term cumulative reward.

The main contribution of this paper is fourfold:

- A novel recommendation model that integrates reinforcement learning with evidential learning to provide uncertainty-aware recommendations.
- Evidence-based uncertainty maximization to enable stability and effective exploration.
- An off-policy formulation to effectively promote the reuse of previously collected data while stabilizing model training, which is important to address data scarcity in recommender systems.
- Seamless integration of a customized RNN, an actor-critic network, and an evidential network to provide an end-to-end integrated training process.

We conduct extensive experiments over four real-world datasets and compare with state-of-the-art baselines to demonstrate the effectiveness of the proposed model.

## 2 RELATED WORK

**Static models.** Matrix Factorization (MF) leverages user and item latent factors to infer user preferences (Koren et al., 2009; Funk, 2006; Koren, 2008). MF is further extended with Bayesian Personalized Ranking (BPR) (Rendle et al., 2012) and Factorization Machine (FM) (Rendle, 2010). Recently, deep learning-based recommender systems (Cheng et al., 2016; Guo et al., 2017) have achieved impressive performance. DeepFM (Guo et al., 2017) integrates traditional FM and deep learning to learn low- and high-order feature interactions. Both wide and deep networks are jointly trained in (Cheng et al., 2016) for better memorization and generalization. In graph-based methods (Berg et al., 2017), users and items are represented as a bipartite graph and links are predicted to provide recommendations. Similarly, Neural Graph Collaborative Filtering (Wang et al., 2019) explicitly encodes the collaborative signal via high-order connectivities in the user-item bipartite graph via embedding propagation.

**Dynamic and sequential models.** Dynamic model shifts latent user preference over time to incorporate temporal information. TimeSVD++ (Koren, 2009) considers time-specific factors, which uses additive bias to model user and item related temporal changes. Gaussian state-space models have been used to introduce time-evolving factors with a one-way Kalman filter (Gultekin & Paisley, 2014). To process implicit data, Sahoo et al. extended the hidden Markov model (Sahoo et al., 2012), and Charlin et al. (Charlin et al., 2015) further augmented it with the Poisson emission. However, these models capture user evolving preference, and they are less aware of future interactions and provide recommendations based on fixed strategies. Similarly, sequential models utilize users' historical interactions to capture users' preferences over time. Tang et al. utilized a CNN architecture to capture union level and point level contributions (Tang & Wang, 2018). Also, Kang et al. leveraged transformer-based user representation to better capture their interest (Kang & McAuley, 2018) and Sun et al. utilized bidirectional encoder for sequential recommendation (Sun et al., 2019). Sequential models neglect long-term users' preferences. The proposed DERL model aims to fill this critical gap by performing evidence guided exploration and maximizing total expected reward.

**RL-based models.** RL-based RS models aim to learn an effective policy to maximize the total expected reward in the long run. The on-policy learning with contextual bandit (Li et al., 2010) and Markov Decision Process (MDP) (Zheng et al., 2018) exploits by interacting with real customers in an online environment. A collaborative contextual bandit algorithm called CoLin (Wu et al., 2016) utilizes graph structure in a collaborative manner. On the other hand, off-policy utilizes Monte Carlo (MC) and temporal-difference (TD) methods to achieve stable and efficient learning with users' history (Farajtabar et al., 2018). Similarly, model-based RL models user-agent interaction via a generative adversarial network (Bai et al., 2019). Pseudo Dyna-Q (Zou et al., 2020) further integrates both direct and indirect RL approaches in a single unified framework without requiring real customer interactions. However, the above methods utilize random exploration strategies, which are less effective at capturing users' long-term preferences. In contrast, our DERL utilizes evidence-based uncertainty to systematically explore the item space to maximize the long-term reward.

## 3 PRELIMINARIES

We first introduce the standard RS setup in RL and provide an overview of evidential theory.

**Recommendation Formulation with RL.** We formulate recommendation tasks in a RL setting, where a RL agent interacts with the environment (users and items) to recommend the next items to a user over time in a sequential order to maximize the cumulative reward. We design this problem as the MDP, which includes a sequence of states, actions, and rewards. More formally, a tuple $(\mathcal{S}, \mathcal{A}, p, r)$ is defined as:

- **State space ($\mathcal{S}$):** A state $\mathbf{s}_t = \text{RNN}(\cdot|\mathbf{s}_{t-1}, \mathbf{u}_t) \in \mathcal{S}$ is generated by a customized RNN that utilizes previous state $\mathbf{s}_{t-1}$ and current user $\mathbf{u}_t$ embedding which is generated from the concatenation of $M$ recently interacted items provided by a sliding window (see details later) and an RL-agent.
- **Action space ($\mathcal{A}$):** An action $\mathbf{a}_t \in \mathcal{A}$ is represented as a continuous parameter vector that recommends top-$N$ items for a user based on the current state $\mathbf{s}_t$ at time $t$.

- **Transition probability** ($p$)**:** The transition probability $p(\mathbf{s}_{t+1}|\mathbf{s}_t, \mathbf{a}_t)$ quantifies the probability from state $\mathbf{s}_t$ to $\mathbf{s}_{t+1}$ with an action $\mathbf{a}_t$.
- **Reward** ($r$)**:** The environment provides an immediate reward as a feedback based on items recommended (actions $\mathbf{a}_t$) to the user in state $\mathbf{s}_t$.

**Uncertainty and the Evidential Theory.** Theory of evidence is a generalization of Bayesian theory to subjective probabilities (Dempster, 1968). We briefly introduce subjective logic (SL) (Jsang, 2016) and discuss uncertainty estimation based on SL. SL is a probabilistic logic that is built upon probability theory and belief theory. It represents uncertainty by introducing vacuity of evidence in its opinion, which is a multinomial random variable $y$ in domain $\mathbb{Y} = \{1, ..., K\}$. This opinion can be equivalently represented by a $K$-dimensional Dirichlet distribution $\mathrm{Dir}(\boldsymbol{p}|\boldsymbol{\alpha})$ where $\boldsymbol{\alpha}$ is a strength over $K$ classes and $\boldsymbol{p} = (p_1, ..., p_K)^{\top}$ governs a categorical distribution over $\mathbb{Y}$. The term *evidence* is the measure of the number of supportive observations from data for each class. It has a fixed relationship with the concentration parameter $\boldsymbol{\alpha}$ given a non-informative prior. Let $e_k$ be the evidence for a class $k$. SL measures different types of second-order uncertainty through evidences, including vacuity, dissonance, and a few others (Josang et al., 2018). In particular, vacuity corresponds to the uncertainty mass of a subjective opinion $\omega$:

$$vac(\omega) = \mathcal{U} = \frac{K}{S}, \quad S = \sum_{k=1}^{K}(e_k + 1) \tag{1}$$

Since vacuity is defined by lack of evidence in the data sample, it provides a natural way to facilitate the exploration of an RL agent, which will be detailed next.

## 4 DEEP EVIDENTIAL REINFORCEMENT LEARNING FOR DYNAMIC RECOMMENDATION

**Overview.** We propose a deep evidential reinforcement learning model to perform dynamic recommendations as shown in Figure 2. The model includes a recurrent neural network (RNN) to maintain dynamic state space, and an evidential-actor-critic (EAC) module to explore the item space by introducing the evidence-based uncertainty (vacuity) into a new evidential RL setting. By incorporating previous state information, recent items captured by a sliding window, and the recommended items from the RL agent, the RNN module generates the current state $\mathbf{s}_t$. This state is further passed to the action network that predicts the mean and variance to form a Gaussian policy distribution. We sample a current action $\mathbf{a}_t$ from the policy distribu-

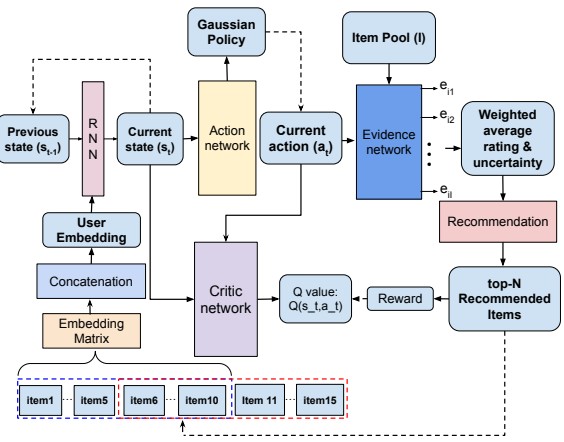

Figure 2: Overview of the DERL framework

tion that corresponds to the latent preference of the user that simultaneously captures the past (via a previous state ), current (through a sliding window) and future interest (through RL exploration). By leveraging the current action and item embeddings, the evidence network provides the evidence that can be used to form the rating prediction for exploitation while estimating the uncertainty for effective exploration. The Q-network (critic) generates an evidential Q-value for evidential policy updates of the action network. Table 4 in Appendix A summarizes the major notations.

### 4.1 ENVIRONMENT SETUP

We start by describing the environment of the proposed evidential RL agent. The environment consists of user-interacted items (an item pool $\mathcal{I}$) from this user's interaction history $H_u$, embedding matrix $E$ to generate the user embedding, the RNN for dynamic state generation, and an evidential reward process (ERP) that specifies an incentive mechanism to each action of the agent. Our reward process encourages a balance between exploitation (based on predicted ratings) and exploration (based on evidence-based uncertainty) when making recommendations to users. In particular, a

recommended list consists of a limited number of items. The proposed ERP ranks candidate items according to an evidential score that integrates the predicted rating and evidence-based uncertainty:

$$\text{score}_{u,i} = \widehat{\text{rating}}_{u,i} + \lambda \mathcal{U}_{\pi}(i|\mathbf{s}_t, \mathbf{a}_t) \tag{2}$$

where $\lambda$ balances the rating and the uncertainty, and $\widehat{\text{rating}}_{u,i}$ is the predicted rating. Given $K$ possible rating classes, the evidence network (introduced later in this section) outputs an evidence vector $\mathbf{e}_i = (e_{i1}, ..., e_{iK})^{\top}$ for each item $i$. This will allow us to evaluate $\widehat{\text{rating}}_{u,i}$ as $\sum_{k=1}^{K} p_{ik} \times k$ where $p_{ik}$ is rating probability given by (9). Meanwhile, uncertainty $\mathcal{U}_{\pi}(i|\mathbf{s}_t, \mathbf{a}_t)$ for item $i$ can be evaluated through (1). Based on the evidential score, an RL agent will choose the top-$N$ items to form a list $\mathcal{N}_u$ and recommend them to the user. As the feedback to the agent, the user provides the actual rating for each recommended items. Consequently, the evidential reward is

$$r_{\pi}^e(\mathbf{s}_t, \mathbf{a}_t) = \frac{1}{N} \left( \sum_{i \in \mathcal{N}_u} (\text{rating}_{u,i} - \tau) + \lambda \mathcal{U}_{\pi}(i|\mathbf{s}_t, \mathbf{a}_t) \right) \tag{3}$$

where $\text{rating}_{u,i}$ is the user assigned ground truth rating and $\tau$ is a threshold chosen based on the rating mechanism ($\tau = 3$ for a $1 - 5$ rating system). Given the evidential reward, we introduce an evidential Q-value, which can be computed by repeatedly applying the Bellman operator ($B^{\pi}$):

$$B^{\pi} Q^e(\mathbf{s}_t, \mathbf{a}_t) \triangleq r_{\pi}^e(\mathbf{s}_t, \mathbf{a}_t) + \gamma \mathbb{E}_{\mathbf{s}_{t+1} \sim \pi}[V(\mathbf{s}_{t+1})] \tag{4}$$

where $V(\mathbf{s}_t) = \mathbb{E}_{\mathbf{a}_t \sim \pi}[Q^e(\mathbf{s}_t, \mathbf{a}_t)]$. The evidential Q-value will be used for the update of EAC module, which is introduced later in this section.

## 4.2 THE CUSTOMIZED RNN FOR LATENT STATE GENERATION

A specially designed RNN is used to maintain the state space of a dynamic RS environment. In particular, a state $\mathbf{s}_t$ is generated by aggregating three pieces of information: the previous state $\mathbf{s}_{t-1}$, items interacted by the user in the the current step, and newly recommended items. Here, an *item* is also an embedding vector which encodes item entity information. By aggregating all this information, the current state can evolve from the previous state by effectively capturing the past preference and future predicted preference of the user. In particular, newly interacted items are extracted from the user's interaction history $H_u$ using a sliding window and the currently recommended items are obtained by invoking the $\mathbf{a}_{t-1}$. Assume that a total $M$ items are obtained with a half from the sliding window and the rest from the action. These $M$ items then go through an embedding matrix to produce a user embedding $\mathbf{u}_t$ for time step $t$. Then, $\mathbf{s}_t$ is formed by

$$\mathbf{s}_t = \text{RNN}(\mathbf{s}_{t-1}, \mathbf{u}_t) \tag{5}$$

To train the customized RNN, we collect additional data tuples $[\mathbf{s}_{t-1}, \mathbf{u}_t]$ into the replay buffer. We then sample batches from the buffer and send $\mathbf{u}_t$ and $\mathbf{s}_{t-1}$ to the RNN module that generates the current state $\mathbf{s}_t$. After that, we send $\mathbf{s}_t$ to action network that samples $\mathbf{a}_t$ from the action distribution. Action $\mathbf{a}_t$ will then go through the evidence network to predict the evidence vector for each candidate item. Finally, we compute evidential loss $J_{\text{Evi}}$ as defined in (10) and conduct backpropagation with respect to RNN parameter $\omega$:

$$\nabla_{\omega} J_{\text{RNN}}(\omega) = \nabla_{\omega} J_{Evi}(\psi) \tag{6}$$

In this way, the computing graph is maintained even in the offline setting and the RNN can be trained as in the standard supervised setting.

## 4.3 EVIDENTIAL ACTOR CRITIC (EAC)

**Training goal.** A standard RL model maximizes the expected sum of rewards. We consider a generalized evidential reward function $r_{\pi}^e$ defined in (3), which augments the standard RL objective with the average evidence-based uncertainty of the recommended items to encourage exploration of the item space. We achieve our training goal by updating the evidential actor network that finds the optimal policy to maximize the expected cumulative evidential reward as:

$$J_{\pi} = \sum_{t=0}^{T} \mathbb{E}_{(\mathbf{s}_t, \mathbf{a}_t) \sim D}(r_{\pi}^e(\mathbf{s}_t, \mathbf{a}_t)) \tag{7}$$

where $D$ is the distribution of $(\mathbf{s}_t, \mathbf{a}_t)$ from the data or the replay buffer and $T$ is the total number of time steps in the episode. A novel benefit of the new objective is to allow the agent to interact with more informative items for more effective exploration of a large item space. EAC consists of three key networks: *action network*, *evidence network*, and *critic network*, which will be detailed next.

**Action network.** The action network (or policy network) utilizes the current state $\mathbf{s}_t$ from the offline replay buffer and outputs a policy distribution $\pi(.|\mathbf{s}_t)$, which is modeled as a Gaussian. From this distribution, we sample an action $\mathbf{a}_t$ that is used in the evidence and the critic networks to provide recommendations and direct the policy update, respectively. For action network update, we use backward update signals from the critic network:

$$\nabla_\phi J_\pi(\phi) = (-\nabla_{\mathbf{a}_t} Q^e(\mathbf{s}_t, \mathbf{a}_t)) \times \nabla_\phi \pi(\cdot|\mathbf{s}_t, \phi) \tag{8}$$

This gradient extends the DDPG style policy update (Lillicrap et al., 2015) by utilizing the chain rule to the Q-network that updates the action network.

**Evidence network.** The evidence network predicts a Dirichlet distribution of class probabilities, which can be considered as an evidence collection process. The learned evidence is informative to quantify the predictive uncertainty of recommended items. The network takes action $\mathbf{a}_t$ from the replay buffer and item pool $\mathcal{I}$ to provide class level evidence. Then, the probability of rating $k$ is

$$p_{ik} = \frac{(e_{ik} + 1)}{S_i} \tag{9}$$

where $e_{ik}$ is the evidence collected for rating $k$ for item $i$. To train the evidence network, we define a standard evidential loss by utilizing the MSE loss between rating class probability $p_{ik}$ and the one-hot ground truth label $\mathbf{y}_i$, in which $y_{ik} = 1$ if $k$ is the correct rating, otherwise $y_{ik} = 0$:

$$J_{Evi}(\psi) = \sum_{k=1}^{K} (y_{ik} - p_{ik})^2 + \frac{p_{ik}(1 - p_{ik})}{S_i + 1} \tag{10}$$

We update the network by backpropagating the evidential loss $J_{Evi}(\psi)$ with its parameters $\psi$.

**Critic network.** The critic network is designed to approximate evidential Q value utilizing the current state $\mathbf{s}_t$ and action $\mathbf{a}_t$ in a fully connected neural network $Q_\theta(\mathbf{s}_t, \mathbf{a}_t)$. This Q-value judges whether the agent generated actions matches the current state $\mathbf{s}_t$ requirements. We derived an update formulation for the critic network following the recent double DQN-based method (Mnih et al., 2015) that utilizes two critic networks to stabilize training process, achieve faster convergence, and provide a better Q-value as:

$$\tilde{Q}^e(\mathbf{s}_t, \mathbf{a}_t) = \mathbb{E}_{\mathbf{s}_{t+1} \sim D, \mathbf{a}_{t+1} \sim \pi}[r_\pi^e(\mathbf{s}_t, \mathbf{a}_t) + \gamma \times \min\{Q^e(\mathbf{s}_{t+1}, \mathbf{a}_{t+1}), \hat{Q}^e(\mathbf{s}_{t+1}, \mathbf{a}_{t+1})\}] \tag{11}$$

where $\hat{Q}(\mathbf{s}_{t+1}, \mathbf{a}_{t+1})$ is a target network, which is updated slowly to stabilize the training process.

The *evidential Q-function* parameters are trained by minimizing the temporal difference (TD) error:

$$J_Q(\theta) = \mathbb{E}_{(\mathbf{s}_t, \mathbf{a}_t, \mathbf{s}_{t+1}, \mathbf{a}_{t+1}, r_\pi^e(\mathbf{s}_t, \mathbf{a}_t)) \sim D} \left[ \frac{1}{2} \left( Q^e(\mathbf{s}_t, \mathbf{a}_t) - \tilde{Q}^e(\mathbf{s}_t, \mathbf{a}_t) \right)^2 \right] \tag{12}$$

where $D$ is the distribution of $(\mathbf{s}_t, \mathbf{a}_t, \mathbf{s}_{t+1}, \mathbf{a}_{t+1}, r_\pi^e(\mathbf{s}_t, \mathbf{a}_t))$ in an offline buffer.

Furthermore, the Q-network is optimized with stochastic gradient decent. The overall recommendation algorithm is shown in Algorithm 1 of Appendix C.

## 4.4 Derivation of Evidential Policy Iteration

We derive evidential policy iteration as a general method for learning optimal uncertainty policies by alternating between evidential policy evaluation and evidential policy improvement in the maximum uncertainty framework. We compute the value of a policy $\pi$ according to the maximum uncertainty objective of Eq. (7). DERL expresses a policy as a Gaussian distribution with mean and covariance of an action neural network. With the above settings, we show that the evidential policy iteration can achieve the optimal policy at convergence.

**Lemma 1** (Evidential Policy Evaluation). *Given the Bellman operator $B^\pi$ in Eq. (4) and $Q^{n+1} = B^\pi Q^n$, the Q-value will converge to the evidential Q-value of policy $\pi$ as $n \to \infty$.*

**Lemma 2** (Evidential Policy Improvement). *Given a new policy $\pi_{new}$ that is updated via Eq (8), then $Q^e_{\pi_{new}}(\mathbf{s}_t, \mathbf{a}_t) \geq Q^e_{\pi_{old}}(\mathbf{s}_t, \mathbf{a}_t)$ for all $(\mathbf{s}_t, \mathbf{a}_t)$.*

**Theorem 3** (Evidential Policy Iteration). *Alternating between evidential policy evaluation and evidential policy improvement for any policy $\pi \in \Pi$ converges to an optimum evidential policy $\pi^*$ such that $Q^{\pi^*}(\mathbf{s}_t, \mathbf{a}_t) \geq Q^e_\pi(\mathbf{s}_t, \mathbf{a}_t)$ for all $(\mathbf{s}_t, \mathbf{a}_t)$.*

Table 2: Performance of Recommendation (average P@N and nDCG@N)

| Category | Model | MovieLens-1M | | MovieLens-100K | | Netflix | | Yahoo! Music | |
|---|---|---|---|---|---|---|---|---|---|
| | | P@5 | nDCG@5 | P@5 | nDCG@5 | P@5 | nDCG@5 | P@5 | nDCG@5 |
| Dynamic MF | timeSVD++ | 0.5341 | 0.4328 | 0.5034 | 0.4145 | 0.5234 | 0.4220 | 0.5267 | 0.4190 |
| | CKF | 0.5567 | 0.4481 | 0.5285 | 0.4322 | 0.5456 | 0.4344 | 0.5344 | 0.4216 |
| Sequential | CASER | 0.5762 | 0.4613 | 0.5434 | 0.4428 | 0.5633 | 0.4542 | 0.5745 | 0.4365 |
| | SASRec | 0.6058 | 0.4862 | 0.5624 | 0.4515 | 0.5958 | 0.4621 | 0.5826 | 0.4422 |
| | BERT4Rec | 0.6122 | 0.4957 | 0.5834 | 0.4855 | 0.5996 | 0.4667 | 0.5901 | 0.4522 |
| Reinforce | $\epsilon$-greedy | 0.5977 | 0.4834 | 0.5580 | 0.4556 | 0.5850 | 0.4765 | 0.5909 | 0.4812 |
| | DRN | 0.6057 | 0.5199 | 0.6154 | 0.5268 | 0.5826 | 0.4720 | 0.6085 | 0.5121 |
| | LIRD | 0.6238 | 0.5332 | 0.6137 | 0.5222 | 0.6134 | 0.5214 | 0.6193 | 0.5238 |
| | CoLin | 0.6162 | 0.5216 | 0.6247 | 0.5285 | 0.5869 | 0.4782 | 0.6112 | 0.5194 |
| Proposed | **DERL** | **0.6313** | **0.5365** | **0.6379** | **0.5386** | **0.6336** | **0.5372** | **0.6232** | **0.5330** |

Please refer to Appendices B.1, B.2 and B.3 for proofs.

**Remark:** The novel of use of vacuity, which is an evidence-based second-order uncertainty, for exploration in RL, can effectively identify uncertain and informative items (from large item space), indicative of users' long-term interest. In particular, the proposed evidential reward encourages the RL agent to recommend items that the model has the least knowledge (as indicated by a high vacuity). After collecting the user feedback, the RL agent can most effectively gain the knowledge on the user preference to make better recommendations in the long run. It should be noted that maximum entropy-based exploration, such as soft-actor-critic (SAC) (Haarnoja et al., 2018), may not reach an optimum policy. It has been shown that a high entropy may imply either high vacuity (lack of evidence) or high dissonance (conflict of strong evidence) (Shi et al., 2020). However, dissonance is not effective for exploration in RS due to its focus on confusing items mostly derived based on the users' current interest. Lemma 2 shows that the evidential reward results in the evidential Q-value that is optimal for the policy improvement. We have also experimentally shown this in a qualitative study by demonstrating better recommendation performance than SAC based exploration.

## 5 EXPERIMENTS

We conduct extensive experiments on four real-world datasets that contain explicit ratings: *Movielens-1M*, *Movielens-100K*, *Netflix*, and *Yahoo! Music*. For baseline comparisons, we use **dynamic** models: timeSVD++ (Koren, 2009), CKF (Gultekin & Paisley, 2014); **sequential** models: CASER (Tang & Wang, 2018), SASRec (Kang & McAuley, 2018), BERT4Rec (Sun et al., 2019); and **RL-based** models: $\epsilon$-greedy (Zhao et al., 2013), DRN (Zheng et al., 2018), LIRD (Zhao et al., 2017), CoLin (Wu et al., 2016). We evaluate rewards based on available ground-truth ratings, which avoids the model from learning from simulated rewards for the non-interacted items that may lead to ineffective recommendations. Further details about datasets, experimental setting, and baseline models are provided in Appendices D, E, and F.

**Evaluation metrics.** We use two standard metrics to measure the recommendation performance. We also use cumulative rewards for the RL-based methods.

- **Precision@N**: It is the fraction of the top-$N$ items recommended in each step of the episode that are positive (rating $> \tau$) to the user. We average over all test users as the final precision.
- **nDCG@N**: Normalized Discounted Cumulative Gain (nDCG) measures ranking quality, considering the relevant items within the top-$N$ of the ranking list in each step of the RL episode.
- **Cumulative Reward**: It measures average reward considering rewards of top-$N$ recommended items in each step for the RL episode.

### 5.1 RECOMMENDATION PERFORMANCE COMPARISON

Table 2 summarizes the recommendation performance from all models. The proposed model benefits from both the RNN module and EAC module so that it provides better results in all datasets. The dynamic and sequential models achieve less ideal performance due to their focus on short-term user interest and inability to provide long-run or future preference. RL methods have shown a clear advantage due to their focus on maximizing expected long-term rewards. Thanks to the evidence-based uncertainty exploration, DERL achieves the best performance among all DL based models.

We further show the step-wise performance of both precision@5 (P@5) and nDCG@5 metrics considering top-5 recommended items in all datasets as shown in Figure 3. We show the average precision and nDCG of the test users over each step after the model is fully trained to demonstrate the

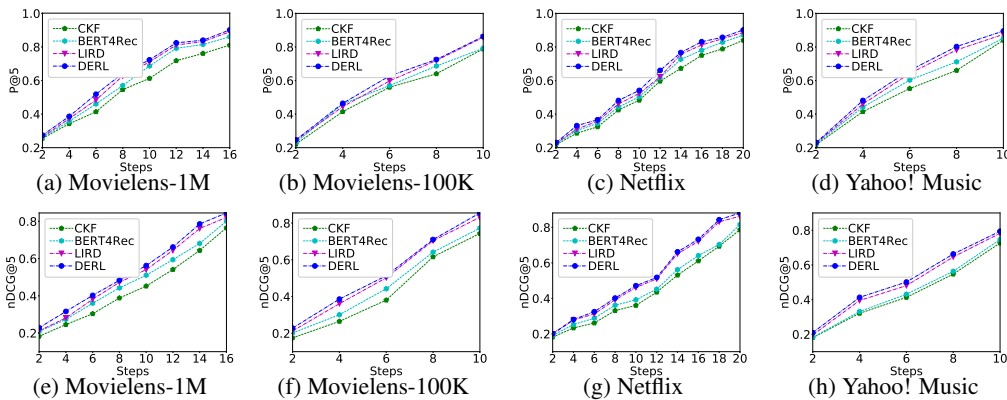

Figure 3: Performance comparison in each time step: (a)-(d): P@5; (e)-(h): nDCG@5

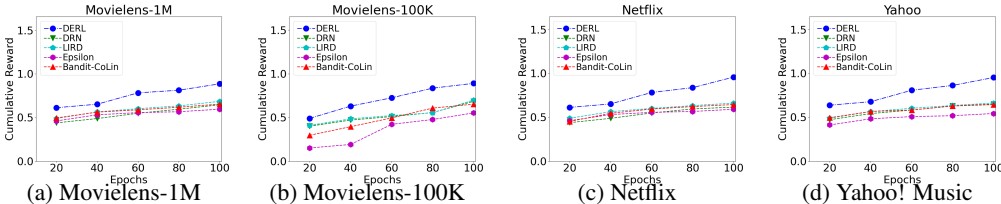

Figure 4: Average cumulative reward for DERL and other RL-baselines in different training stages

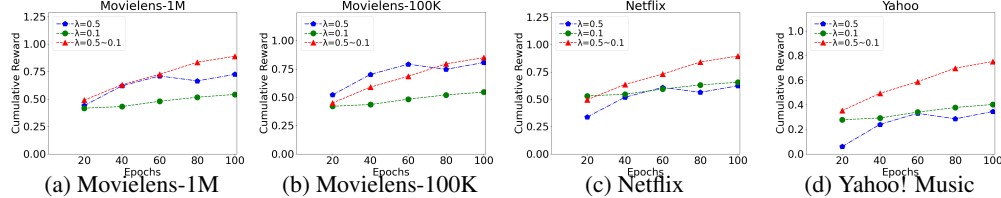

Figure 5: Average cumulative reward for DERL for different $\lambda$

effectiveness of the dynamic recommendation. We fixed the step size to 16, 10, 20, and 10 for the Movielens-1M, Movielens-100K, Netflix, and Yahoo! Music datasets based on their average number of user-item interactions, respectively. At the initial steps, both precision and nDCG are low for all models (we choose the best model from each category as shown in Table 2). This is as expected due to lack of user interactions. All the models start to improve after the initial stage. Dynamic models and sequential models still have poor performance compared to the RL-based methods. The proposed DERL model provides consistently better performance over the entire process. However, it has a smaller advantage at the beginning due to its strong focus on exploration. It is also worth to note that the difference between DERL and other models appears to be smaller on the plots because of the wide range of the $y$-values ($0.2 - 1$ in most cases) to cover the entire recommendation cycle.

## 5.2 ABLATION STUDY

First, we provide a comparison of the average cumulative reward to demonstrate how the proposed model achieves higher cumulative reward than other RL-based models. Second, we analyze the impact of hyperparameter ($\lambda$) that balances exploitation and exploration in the proposed model.

**Cumulative reward for RL-based methods.** We consider cumulative reward to measure test users' recommendation performance. We plot the average cumulative rewards for the proposed DERL model and baseline RL-models in Figure 4 to show their average reward over different training epochs in all four datasets. As can be seen, the cumulative rewards for DERL and RL-based model in the initial epochs are quite close. But in later epochs, DERL clearly outperforms the other baselines. This is because the model explores more effectively during the training process to enhance the knowledge of the model. Further, reward gains for those baselines are largely similar. But for DERL, it has shown a significant improvement in comparison with those baselines.

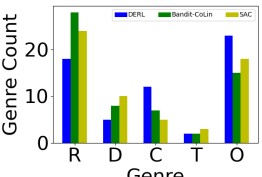 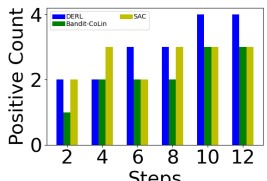 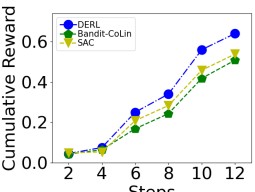

Figure 6: Genre count, Positive count, cumulative reward for a given user

**Impact of hyperparameter ($\lambda$).** The hyperparameter ($\lambda$) plays a critical role in recommending the top-$N$ items and generating the evidential reward. We test three different settings: $\lambda = 0.1$, $\lambda = 0.5$, and gradually reducing $\lambda$ from 0.5 to 0.1. As can be seen from Figure 5, dynamically adjusting $\lambda$ achieves consistently better performance on all datasets. This supports the intuition that in the early steps, a large $\lambda$ allows the model to conduct sufficient exploration. Once the model gains sufficient knowledge from the environment and is able to make accurate recommendation, reducing $\lambda$ will allow the model to exploit its knowledge to provide effective recommendations.

## 5.3 QUALITATIVE STUDY

We conduct a qualitative analysis to show the advantage of using evidence-based uncertainty (vacuity) for RL exploration than other two competitive baselines: entropy-guided exploration as in the soft actor-critic (SAC) (Haarnoja et al., 2018) and a contextual bandit algorithm (CoLin) (Wu et al., 2016). We select a random test user (ID:4967) from the Movielens-1M dataset and

Table 3: Recommend movies for UserID: 4967

| Model | Movies | Movie Genre | Vacuity |
|---|---|---|---|
| DERL | Kids of the Round Table (1995) | **Adventure,Fantasy** | 0.12 |
| | Postino, Il (The Postman) (1994) | **Romance** | 0.11 |
| | How to Make an American Quilt (1995) | **Drama** | 0.14 |
| | Pocahontas (1995) | **Musical** | 0.12 |
| | Three Lives and Only One Death (1996) | Comedy | 0.22 |
| SAC | Private Benjamin (1980) | Comedy | 0.11 |
| | Return of the Pink Panther, The (1974) | Comedy | 0.12 |
| | Lawnmower Man 2: Beyond Cyberspace (1996) | **Sci-Fi,Thriller** | 0.09 |
| | Ruling Class, The (1972) | Comedy | 0.11 |
| | Love in Bloom (1935) | **Romance** | 0.09 |
| CoLin | Young Sherlock Holmes (1985) | **Adventure** | 0.11 |
| | Karate Kid, Part II, The (1986) | **Fantasy** | 0.07 |
| | Mighty Joe Young (1998) | **Adventure** | 0.08 |
| | Christmas Vacation (1989) | Comedy | 0.10 |
| | Father of the Bride Part II (1995) | Comedy | 0.12 |

show the genre counts, positive counts of recommended items, and cumulative reward in each step as shown in Figure 6. At the initial few steps, SAC has more positive counts but is less effective in exploration. DERL is able to explore more informative items (evidenced by more diverse genres), where different genres are denoted as Romance (R), Drama (D), Comedy (C), Thriller (T), Others (O) in the left plot. In later steps, DERL consistently outperforms both competitive model due to the better utilization of evidence-based uncertainty to discover more informative future items which could reflect the user's long-term preference. Table 3 shows the predicted vacuity for each recommended item. The overall higher vacuity scores indicate that DERL recommends more items that are currently unknown to the users, which is instrumental to explore their long-term interests. It also explores more diverse genres (5 vs. 3) of items than the baselines. The results show a consistent trend: DERL focuses on exploration in the earlier phase by recommending more diverse items (left plot) and both competitive methods achieve better performance (more positive counts) during this phase (middle plot). Due to better exploration, DERL eventually achieves a much better cumulative reward in the later phase (right plot). More specifically, as Table 3 shows, in early steps, DERL identifies four out of five important items (genre types in bold) that are recommended based on (estimated) future long-term interest. However, SAC finds two important movies based on genre and three by the CoLin. SAC selects three comedy movies, which only reflects the user's current preference from the current and past interactions. Similarly, CoLin is also more focused on comedy rather than exploring diverse movies.

## 6 CONCLUSION

In this paper, we propose a novel deep evidential reinforcement learning framework for dynamic recommendations. The proposed DERL framework learns a more effective recommendation policy by integrating both the expected reward and evidence-based uncertainty. DERL integrates a customized RNN to generate the current state that accurately captures user interest and an evidential-actor-critic module to perform evidence-based exploration to optimize policy by improving an evidential Q-value. We theoretically prove the convergence behavior of the proposed evidential policy integration strategy. Experimental results on real-world data and comparison with the state-of-the-art competitive models demonstrate the effectiveness of the proposed model.

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

# Appendix

**Organization of Appendix.** In this Appendix, we first summarize the major mathematical notations in Appendix A. We then present the proofs of lemmas and theorems in Appendix B. We show the detailed DERL algorithm in Appendix C. We present the details of the dataset in Appendix D, experimental setting in Appendix E, and baseline models in Appendix F. Further we include some additional comparison results in Appendix G. The link to the source code is given in Appendix H.

## A  SUMMARY OF NOTATIONS

Table 4: Summary of Notations

| Notation | Description |
|---|---|
| $T$ | step size or length of episode |
| $u, i$ | user (episode) and item indices |
| $E$ | embedding matrix for item embedding |
| $\mathbf{u}_t$ | user $u$'s embedding at time step t |
| $\mathbf{s}_t, \mathbf{a}_t$ | state and action at time t |
| $e_{ik}, p_{ik}$ | evidence and evidence-based probability on rating class $k$ for item $i$ |
| $y_i$ | one-hot rating label on item $i$ |
| $\phi, \psi$ | parameters of action and evidence networks |
| $\theta, \omega$ | parameters of critic and recurrent networks |
| $\pi, Q(\mathbf{s}_t, \mathbf{a}_t)$ | recommendation policy or action network, Q value function or critic network |
| $\widehat{\text{rating}}_{u,i}, \text{rating}_{u,i}$ | predicted and actual rating for user $u$ on item $i$ |
| $\text{score}_{u,i}$ | evidential score for user $u$ on item $i$ |
| $\tau, \lambda$ | rating threshold and balance hyper-parameter for exploitation and exploration |
| $\mathcal{U}_\pi(i\|\mathbf{s}_t, \mathbf{a}_t)$ | item $i$'s evidence-based uncertainty |
| $r_\pi^e(\mathbf{s}_t, \mathbf{a}_t), Q^e(\mathbf{s}_t, \mathbf{a}_t)$ | evidential reward and Q value |
| $K, M$ | the number of rating class, the number of interacted items in each time step |
| $\mathcal{N}_u$ | top-$N$ recommended items list for user $u$ |
| $H_u, \mathcal{I}$ | user $u$'s interaction history and item pool |

## B  PROOFS OF THEORETICAL RESULTS

In this section, we provide proofs of all lemmas and the theorem.

### B.1  PROOF OF LEMMA 1

Given the evidential reward defined as $r_\pi^e(\mathbf{s}_t, \mathbf{a}_t) = \frac{1}{N}\left(\sum_{i \in \mathcal{N}_u}(\text{rating}_{u,i} - \tau) + \lambda \mathcal{U}_\pi(i|\mathbf{s}_t, \mathbf{a}_t)\right)$ the update rule for evidential Q-value can be written as:

$$Q^e(\mathbf{s}_t, \mathbf{a}_t) = \mathbb{E}_\pi \sum_{t'=t}^{\infty} \gamma^{t'} r_\pi^e(\mathbf{s}_{t'}, \mathbf{a}_{t'}) = r_\pi^e(\mathbf{s}_t, \mathbf{a}_t) + \gamma \mathbb{E}_{\mathbf{s}_{t+1}, \mathbf{a}_{t+1}}[Q^e(\mathbf{s}_{t+1}, \mathbf{a}_{t+1})] \tag{13}$$

Then based on the evaluation convergence rule (Sutton et al., 1999) with finite action space, it is guaranteed that the Q-value will converge to the evidential Q-value of policy $\pi$.

### B.2  PROOF OF LEMMA 2

The policy can be updated towards the new Q-value function. Consider the updated policy $\pi_{new}$ as the optimizer of the maximization problem.

$$\pi_{new} = \arg\max_{\pi'} J_\pi(\phi) = \arg\max_{\pi'} \mathbb{E}_{\mathbf{s}_t \sim D, \mathbf{a}_t \sim \pi'}[Q_{\pi'}^e(\mathbf{s}_t, \mathbf{a}_t)] \tag{14}$$

Denote the old policy as $\pi_{old}$. Using the update rule specified in Eq (8) with a sufficiently small step size, we get an updated policy $\pi_{new}$ that satisfies

$$\mathbb{E}_{\mathbf{a}_t \sim \pi_{new}}[Q^e_{\pi_{old}}(\mathbf{s}_t, \mathbf{a}_t)] \geq \mathbb{E}_{\mathbf{a}_t \sim \pi_{old}}[Q^e_{\pi_{old}}(\mathbf{s}_t, \mathbf{a}_t)] \tag{15}$$

Given Eq (15), we have the following inequality

$$\begin{aligned} Q^e_{\pi_{old}}(\mathbf{s}_t, \mathbf{a}_t) \leq &r^e(\mathbf{s}_t, \mathbf{a}_t) + \gamma \mathbb{E}_{\mathbf{s}_{t+1}, \mathbf{a}_{t+1} \sim \pi_{new}}[Q^e_{\pi_{old}}(\mathbf{s}_{t+1}, \mathbf{a}_{t+1})] \\ \leq &r^e(\mathbf{s}_t, \mathbf{a}_t) + \gamma \mathbb{E}_{\mathbf{s}_{t+1}, \mathbf{a}_{t+1} \sim \pi_{new}}[r^e(\mathbf{s}_{t+1}, \mathbf{a}_{t+1})] \\ &+ \mathbb{E}_{\mathbf{s}_{t+2}, \mathbf{a}_{t+2} \sim \pi_{new}}[Q^e_{\pi_{old}}(\mathbf{s}_{t+2}, \mathbf{a}_{t+2})] \\ & \cdots \\ = &Q^e_{\pi_{new}}(\mathbf{s}_t, \mathbf{a}_t) \end{aligned} \tag{16}$$

where $r^e(\mathbf{s}_t, \mathbf{a}_t)$ is a evidential reward in step $t$. Therefore, we show that the new policy $\pi_{new}$ ensures $Q^e_{\pi_{new}}(\mathbf{s}_t, \mathbf{a}_t) \geq Q^e_{\pi_{old}}(\mathbf{s}_t, \mathbf{a}_t)$ for all $(\mathbf{s}_t, \mathbf{a}_t)$.

### B.3 PROOF OF THEOREM 1

Let $\pi_i$ denote the policy at iteration $i$. We already show that the sequence $Q^e_{\pi_i}(\mathbf{s}_t, \mathbf{a}_t)$ is monotonically increasing. Since $Q^e_{\pi}(\mathbf{s}_t, \mathbf{a}_t)$ is bounded above, the sequence converges to some $\pi^*$. At convergence, it must be the case that $J_{\pi^*}(\pi^*(.|\mathbf{s}_t)) \leq J_{\pi^*}(\pi(.|\mathbf{s}_t))$ for $\pi \neq \pi^*$. Based on Lamma 2, we have $Q^e_{\pi^*}(\mathbf{s}_t, \mathbf{a}_t) > Q^e_{\pi}(\mathbf{s}_t, \mathbf{a}_t)$ for all $(\mathbf{s}_t, \mathbf{a}_t)$. In other words, the evidence value of any other policy $\pi$ is lower than that of the converged policy $\pi^*$. Therefore, it guarantees convergency to an optimal policy $\pi^*$ such that:

$$Q^e_{\pi^*}(\mathbf{s}_t, \mathbf{a}_t) \geq Q^e_{\pi}(\mathbf{s}_t, \mathbf{a}_t) \tag{17}$$

## C DEEP EVIDENTIAL REINFORCEMENT LEARNING ALGORITHM

---
**Algorithm 1** Deep Evidential Reinforcement Learning
---
**Require:** Hyperparameters: $\alpha, \beta, \lambda, \tau$, and time step: $T$
1: Initialize RNN: $\omega$, action network: $\phi$, evidence network: $\psi$, and critic network: $\theta$ , initial state: $\mathbf{s}_0$ and initial user embedding: $\mathbf{u}_0$ with $M$ items from interactino history $H_u$
2: **for** each epoch **do**
3:     **for** each user as an episode **do**
4:         **for** $t \in T$ **do**
5:             Compute state: $\mathbf{s}_t$ with (5).
6:             Compute action: $\mathbf{a}_t \sim \pi_\theta(.|\mathbf{s}_t)$
7:             Compute evidential score using (2)
8:             Recommend $top\text{-}N$ items based on computed evidential score.
9:             Compute rewards for top-$N$ items utilizing Equation 3.
10:            Add $(u_t, \mathbf{s}_{t-1}, r^e_\pi(\mathbf{s}_t, \mathbf{a}_t), \mathcal{U}_\pi(.|\mathbf{s}_t, \mathbf{a}_t), done)$ into replay buffer
11:            Take $\frac{M}{2}$ items from the sliding window and other $\frac{M}{2}$ items from RL-agent recommended items
12:            **for** each gradient step **do**
13:                 Sample batched data from replay buffer and update:
14:                 Update action network with (8)
15:                 Update evidence network with (10)
16:                 Update critic network with (12)
17:                 Update RNN network with (6)
18:            **end for**
19:         **end for**
20:     **end for**
21: **end for**
---

## D DESCRIPTION OF THE DATASETS

We evaluated DERL on four public benchmark datasets that contain explicit ratings:

- **Movielens-1M**[1]: This dataset includes 1M explicit feedback (ratings) made by 6,040 anonymous users on 3,900 distinct movies from 04/2000 to 02/2003.
- **Movielens-100K**[2]: This dataset contains 100,000 explicit ratings on a scale of (1-5) from 943 users on 1,682 movies. Each user at least rated 20 movies from September 19, 1997 through April 22, 1998.
- **Netflix** (Bennett et al., 2007): This dataset has around 100 million interactions, 480,000 users, and nearly 18,000 movies rated between 1998 to 2005. We pre-processed the dataset and selected 6,042 users with user-item interactions from 01/2002 to 12/2005.
- **Yahoo! Music rating** (Dror et al., 2012): The dataset includes approximately 300,000 user-supplied ratings, and exactly 54,000 ratings for randomly selected songs. The ratings for randomly selected songs were collected between August 22, 2006 and September 7, 2006.

## E   EXPERIMENTAL SETTING

We consider each user an episode for the RL setting and split users into 70% as training users and 30% as test users. For each user, we select the first $M = 10$ interacted items to represent an initial state $s_0$. In the next state, we utilize previous state representation and concatenate five items embedding from sliding window and other five items embedding from RL agent to generate current state $s_t$ by passing through the RNN module. Then, the action network generates mean and covariance for a Gaussian policy from which action is sampled. This action is further passed to the evidence network, which utilizes the embeddings of user interacted items to produce corresponding evidence for each item. We use the setting of classification, where explicit ratings are used as class labels. With that evidence, we compute the evidential score by evaluating evidence-based rating and uncertainty to rank those items, which provides a list of top-$N$ final recommendations. We then evaluate the evidential reward. We put 10 interactions into each period and set $\tau = 3$.

## F   COMPARISON BASELINES

We compare with dynamic, sequential, and reinforcement learning models:

- **Dynamic models** include standard dynamic matrix factorization model timeSVD++ (Koren, 2009) as the time-evolving latent factorization model and collaborative Kalman filtering (CKF) (Gultekin & Paisley, 2014).
- **Sequential models** include Sequential Recommendation via Convolutional Sequence Embedding (Caser) (Tang & Wang, 2018), attention-based sequential recommendation model (SASRec) (Kang & McAuley, 2018), and sequential recommendation with bidirectional encoder (BERT4Rec) (Sun et al., 2019).
- **Reinforcement learning-based models** include $\epsilon$-greedy (Zhao et al., 2013), deep Q-network based news recommendation (DRN) (Zheng et al., 2018), and actor-critic based list-wise recommendation (LIRD) (Zhao et al., 2017), and contextual bandit based method CoLin (Wu et al., 2016).

## G   ADDITIONAL EXPERIMENTS AND COMPARISON RESULTS

In this section, we present additional experiments, with a focus on comparing different types of baselines.

### G.1   COMPARISON WITH CONTEXTUAL BANDIT BASED METHODS

In the main paper, we compared with a state-of-the-art collaborative contextual bandit based recommendation method, Colin (Wu et al., 2016). Here, we include two additional bandit based models [3]:

---

[1] https://grouplens.org/datasets/movielens/1M/
[2] https://grouplens.org/datasets/movielens/100k/
[3] https://github.com/HCDM/BanditLib

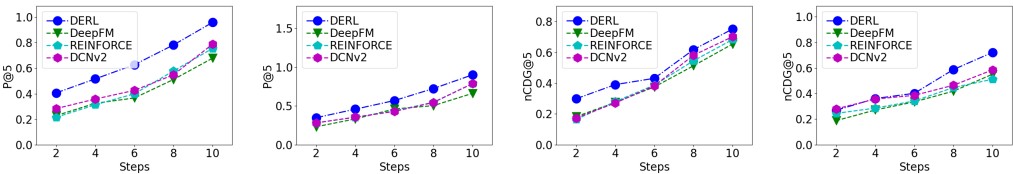

Figure 8: Performance comparison in each time step

LinUCB (Lin) and Hybrid-LinUCB (HLin), to show a more complete comparison. We report the cumulative reward. DERL shows a clear advantage over all three bandit methods, which further justifies its better exploration capability to capture users' long-term interest.

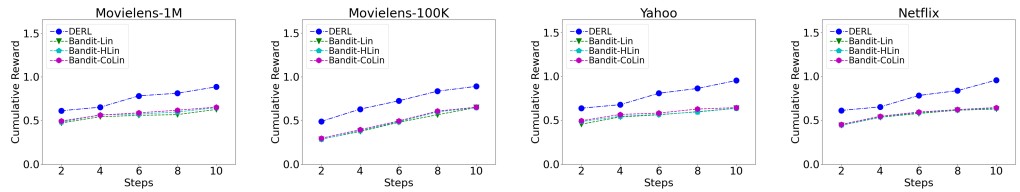

Figure 7: Comparison between bandit-based models and ours in four different datasets.

## G.2 COMPARISON WITH OTHER BASELINES

In this section, we include two recent models for comparison: DeepFM (Guo et al., 2017) and DCNv2 (Wang et al., 2021). DeepFM integrates traditional factorization machine and deep learning to learn low- and high-order feature interactions. Similarly, DCNv2 is more expressive to learn feature interactions and also more cost-efficient. We also include one classical RL-based method called REINFORCE (Chen et al., 2019a), which applies off-policy learning to handle data bias. The test performance metric P@5 and nDCG@5 among the proposed model DERL and above three baselines in two datasets Movielens-1M and Movielens-100K are shown in Table 5 and Figure 8. Although these two deep learning-based recommender models achieve reasonable recommendation performance, they mainly lack to handle the temporal preference of the users and hence perform worse than the proposed DERL method. Furthermore, REINFORCE has limited exploration power and cannot effectively capture long-term user preference in the future, hence its performance is also lower than DERL. We further add a static recommendation model, LightGCN (He et al., 2020), which computes user and item embeddings via a linear aggregation of its neighbors. Its performance is lower than DERL in both datasets because it is ineffective in handling highly sparse user interactions in a dynamic setting. In addition, we also compare with two recent RL-based methods: DRR (Fu et al., 2021) and DHRC (Liu et al., 2020). The significant advantage of exploration over these baselines further confirms its outstanding recommendation performance achieved by DERL.

Table 5: Comparison of Recommendation Performance (average P@N and nDCG@N)

| Model | MovieLens-1M | | MovieLens-100K | |
|---|---|---|---|---|
| | P@5 | nDCG@5 | P@5 | nDCG@5 |
| DeepFM | 0.5647 | 0.4625 | 0.5428 | 0.4514 |
| LightGCN | 0.5668 | 0.4884 | 0.5714 | 0.4852 |
| DRR | 0.5370 | 0.5112 | 0.5890 | 0.5225 |
| DHCRS | 0.4980 | 0.5080 | 0.5214 | 0.4746 |
| REINFORCE | 0.6074 | 0.5116 | 0.5926 | 0.5149 |
| DCNv2 | 0.6152 | 0.5187 | 0.6158 | 0.5166 |
| DERL | **0.6313±0.058** | **0.5365±0.031** | **0.6379±0.061** | **0.5386±0.034** |

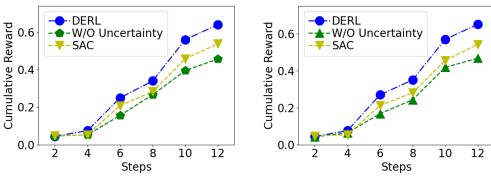

Figure 9: Comparisons exploration strategies in Movielens-1M (left) and Movielens-100K (right).

### G.3 IMPACT OF VACUITY FOR EXPLORATION

Uncertainty is commonly used to support exploration of RL agents in different applications. Evidential deep learning leverages second-order uncertainty (i.e., vacuity) to perform effective exploration. The proposed evidential reward encourage the RL agent to recommend items that the model has the least knowledge (as indicated by a high vacuity). After collecting the user feedback, the RL agent can most effectively gain the knowledge on the user preference to make better recommendations in the long run. The effectiveness of vacuity guided exploration has been demonstrated in both the motivating example in the introduction and also our empirical evaluations.

We further investigate the effect of vacuity in our proposed model by comparing DERL with an alternative design without vacuity. Furthermore, we also compare exploration using the first order uncertainty, which is employed by soft-actor-critic (SAC). We show the comparison results on two datasets in Figure 9. It can be seen that without uncertainty guided exploration, the model collects the least cumulative reward in a long run. SAC utilizes entropy based exploration and achieves better cumulative reward than without uncertainty guided exploration. This provides evidence that role of the exploration is crucial in RL-based recommendation. However, it performs worse than the vacuity based DERL method. This is because vacuity guided exploration allows our model to focus its exploration on the most informative items that help the model gain the most knowledge to form an optimal policy. The advantage over entropy-based exploration is also consistent with our earlier discussion in Section 4.4 of of the main paper.

## H LINK FOR THE SOURCE CODE

https://anonymous.4open.science/r/EvidentialRecommendation-2BDE/README.md

