# OpenReview forum: "Deep Evidential Reinforcement Learning for Dynamic Recommendations"
_ICLR.cc/2023/Conference — Submitted to ICLR 2023_

### Official Review · Reviewer_JmPi · 2022-10-23

**Confidence:** 5
**Clarity, Quality, Novelty And Reproducibility:** 1.The paper is well written.

2.The p…
**Correctness:** 2
**Technical Novelty And Significance:** 2
**Empirical Novelty And Significance:** 2
**Recommendation:** 3

**Strength And Weaknesses:**

Strength:

1.Exploration in RL for RS is an important problem.

2.The paper is generally well written and easy to follow.

3.The experiments are on several public datasets.

Weakness:

1.The technical quality of this paper is not enough, and it seems like a direct combination with Evidential Theory and Reinforcement Learning.

2.The paper is not sound as there are many exploration methods in RL literature, such as count-based methods and intrinsic motivations(RND,ICM). But the paper does not discuss and compare these methods.

3.The theoretical analysis is not novel, as it is a direct result of RL theory.

4.The update rule of the critic network does not follow Double DQN, but follow the clipped double q-learning in the well known TD3 algorithm.

5.The paper does not provide a specification of the experimental setup. Did the authors bulid simulator? If not, how to evaluate the performance of each policy in the offline setting?

6.Why not compare SAC in Table 2 as SAC is compared in Figure 6?

7.How to verify that the performance improvement over pervious RL methods indeed comes from the evidential reward? As we can see, you choose some advanced techniques like Eq (11), and it is not deployed in previous baselines.

**Summary Of The Paper:**

This paper proposes a exploration method, called  deep evidential reinforcement learning (DERL)  to improve the exploration ability of RLRS. The DERL algorithm uses recurrent neural network to represent the dynamic feature of the user state, and conducts evidential-actor-critic module to enable better exploration. Experimental results validates its effectiveness over dynamic models, sequential models, and previous RL for RS methods. Finally, some ablation studies and qualitative studies are conducted.

**Summary Of The Review:**

Overall, the paper study an important problem for RL in RS. The paper is well written, and does experiments on several public datasets.
However, the novelty of the paper is limited, and does not discuss and compare current exploration methods. Also, the experiments are not sound. Thus, I choose to reject this paper at this version.

---

> ### Author Response · Authors · 2022-11-18
> **Response to Reviewer JmPi [Part I]**
>
> Thank you for reviewing our paper and providing valuable comments. We summarize our response as follows:
>
> **Q1. Not enough technical quality of the paper.**
>
> To our best knowledge, this is the first work that performs systematic evidence-guided exploration in reinforcement learning for dynamic recommendations. We not only consider the exploration in our framework but also design a customized RNN to capture user dynamics and effectively represent the current state for the RL agent. Similarly, we have designed a novel evidential reward function incorporating the maximum reward RL objective in terms of ratings on the items with evidence-based uncertainty maximization. The evidence-based uncertainty formulation substantially improves exploration and robustness by acquiring diverse items that are indicative of a user's long-term interest.
>
> **Q2. Doesn't discuss and compare with many exploration methods in RL.**
>
> Thank you for the valuable suggestions. We agree that there are various exploration strategies like count-based and intrinsic motivation explorations in RL. However, none of these strategies have been leveraged in recommendation systems and some of them are not applicable to the unique recommendation settings. In the problem setting that we consider, the count-based approach is unsuitable because we utilize continuous state generation via a customized RNN whereas the count-based relies on discrete state. Similarly, intrinsic motivation doesn't consider direct feedback, while our setting uses explicit ratings that provide direct feedback. In fact, in addition to the four RL based recommendation baselines as we compared in the main paper, we introduce REINFORCE as another RL baseline in Appendix G.2. Here, we further compare with two additional recent RL baselines, including DDR [1] and DHCRS [2]. The results further justify the advantage of the proposed DERL model.
>
>
> | **Model**     | **MovieLens-1M**|**MovieLens-1M** |**MovieLens-100K** |**MovieLens-100K** |
> |  -- | -----------  |--|--|--|
> |     | **P@5** |**nDCG@5** |**P@5** |**nDCG@5** |
> | DRR [1] | 0.5370 | 0.5112 | 0.5890 |  0.5225 |
> | DHCRS [2] | 0.4980 | 0.5080 | 0.5214 | 0.4746 |
> |**DERL**|**0.6313$\pm$0.058**|**0.5365$\pm$0.031**|**0.6379$\pm$0.061**|**0.5386$\pm$0.034**|
>
>
> **Q3. Theoretical contribution is not novel.**
>
> We propose a novel policy called evidential policy based on the evidential reward uniquely designed to support effective exploration in the context of recommender systems. While it may not be totally novel, we argue that it is essential to show that the proposed evidential policy iteration can achieve the optimal policy at convergence. To this end, we derive a theorem for our evidential policy iteration based on the classical RL theories to show its convergence property.
>
>
> **Q4. The update rule of the critic network**
>
> Thank you for pointing out this. We clarify that a more recent double DQN-based method [3] is used and we will provide a reference for clarification in the revised paper.
>
>
> **Q5.  Specification of the experimental setup and evaluation of each policy in the offline setting.**
>
> The environment setup for the RL agent and the detailed experimental settings are provided in Section 4.1 and Appendix E, respectively. More specifically, the RL environment consists of user-interacted items (i.e., an item pool $\mathcal{I}$) from the user's interaction history $H_u$ , an embedding matrix $E$ to generate the user embedding, a RNN for dynamic state generation, and an evidential reward process (ERP) that specifies an incentive mechanism to each action of the agent. We consider each user as an episode and split users into 70\% as training users and 30\% as test users. For each user, we select the first $M=10$ interacted items to represent an initial state ${\bf s}_0$. In the next state, we utilize previous state representation and concatenate next five item embeddings from the sliding window and the other five item embeddings from RL agent's recommendation to generate the current state. Our evidential policy network updates its parameter in each training stage and we evaluate each policy through test users until its convergence. We have shown that our converged policy has achieved higher cumulative reward in Figure 4 than existing methods.
>
> **Q6. Why not compare SAC in Table 2?**
>
> Thank you for the suggestion. We clarify that SAC is never implemented in recommender systems. Instead of considering SAC as a baseline, we think it would be more appropriate to show it in the ablation study to demonstrate the effectiveness of evidence-based exploration.

---

> ### Author Response · Authors · 2022-11-18
> **Response to Reviewer JmPi [Part II]**
>
> **Q7. Performance improvement from evidential reward.**
>
> Our evidential reward function considers not only the highest rating items but also the most important items that lead to improved cumulative reward in the long run. In Figure 9 of Appendix G.3, where we investigate the impact of vacuity for exploration, we showed that the DERL with evidential reward outperforms a model without evidential exploration.
>
> **References**
>
> - [1] F. Liu, R. Tang, X, et al. "State representation modeling for deep reinforcement learning based recommendation" Knowl.-Based Syst., vol. 205, p. 106170, October 2020.
>
> - [2] M. Fu et al. Deep Reinforcement Learning Framework for Category Based Item Recommendation. IEEE Transactions on Cybernetics, 2021.
>
> - [3] Mnih, Volodymyr, et al. "Human-level control through deep reinforcement learning." nature 518.7540 (2015): 529-533.

---

### Official Review · Reviewer_h31X · 2022-10-23

**Confidence:** 3
**Correctness:** 3
**Technical Novelty And Significance:** 3
**Empirical Novelty And Significance:** 3
**Recommendation:** 8

**Clarity, Quality, Novelty And Reproducibility:**

The paper is well-written and easy to read. Some important pieces could be explained in more detail to avoid the reader having to guess how things were implemented, but overall I had little trouble understanding the proposed method.
In my opinion, this approach to exploration is novel and aims to address a common issue in recommender systems. I have not seen the concept of vacuity being used as an exploration bonus for RL in recommenders.
There are enough details in the paper to reimplement the architecture, so I believe that reproducibility should not be a major issue.

**Strength And Weaknesses:**

Strength:
- The paper introduces a novel idea for exploration, as far as I am aware. Maintaining varied recommendations, while still showing relevant items to the user is a difficult problem, and the introduction of vacuity seems to be doing a good job at addressing that problem.
- The authors evaluated in a large number of datasets over multiple competing benchmarks, which strengthen confidence in the propose method.



Weakness:
- There are a few key components of the paper that were not completely clear (see summary of review). For example, in eq 1, what constitutes evidence "e_k"? It should take some minor editing to clarify some of these questions.
- I understand it's difficult to do, but since the paper focus is on exploration, it would have strengthened the work quite a bit to have an online experiment. Exploration means producing results that have not been produced before, and these counterfactuals cannot be evaluated from historical data.



**Summary Of The Paper:**

This paper proposes a new approach to exploration in RL for recommender systems. The authors propose using the concept of vacuity to model uncertainty about an item's rating, and include it as an exploration bonus into the reward function.
Offline results through multiple datasets show that the proposed method, DERL, not only is able to achieve higher top-5 precision compared to other methods, but produces more varied results thanks to the exploration component.

**Summary Of The Review:**

The paper proposes a novel approach to exploration for RL in the context of recommender systems.
The authors ran extensive tests in which the results align with the claim made. Most importantly, the qualitative show by inspection that DERL is able to produce more varied results, instead of biasing recommendations to one particular genre.

The paper is well written, but I would ask the authors to clarify a few things in the paper with more details:

1 - In eq 9, what precisely is "e_k" in this case? What constitutes evidence for class k?
2 - Eq 10., why use MSE as a loss of the first component when you have "p_ik" and "y_ik"? Isn't this akin to a classification problem, and you could use cross-entropy?
3 - For the uncertainty "U(i | s, a)", can you clarify how eq. 1 is used in the evaluation. As defined, Eq. 1 is K / sum_k^K (e_k + 1), which means it is not a function of any particular class (K is number of classes, and the bottom is a sum over all classes).
So how can U( i | s ,a ) be evaluated for 1 particular item? Wouldn't  i be considered 1 class over all possible classes?

---

> ### Author Response · Authors · 2022-11-18
> **Response to Reviewer h31X**
>
> Thank you for reviewing our paper and providing valuable comments. We summarize our response as follows:
>
> **Q1. What constitutes evidence $e_k$?**
>
> Thank you for the suggestion. We would like to clarify that evidence $e_k$ is a non-negative number that can be interpreted as the support for the $k$-th class.
>
>
> **Q2. Online experiment for exploration**
>
> Thanks for your thoughtful comment. We would like to mention that we set up the RL environment to best simulate an online setting. For all the testing users, the RL agent is only exposed to the already interacted items, then gets feedback signals and predicts the future interactions. An online experiment as suggested by the reviewer is an important future work to further justify the performance of the proposed approach.
>
>
> **Q3.  Why use MSE loss for a classification problem?**
>
> Under the evidential setting, it can be shown that by minimizing the MSE loss, it simultaneously minimizes the prediction error and the variance of the Dirichlet distribution that generates the evidence. So, it usually leads to better practical performance for evidential learning than other losses.
>
>
> **Q4.  Clarification on Eq.1 for evaluation**
>
> Vacuity in Eq.1 is computed for each item rather than for each class. It captures the agent's overall lack of knowledge on the item.  More specifically, given an item $i$, the evidence network can predict evidence $e_{ik}$ for each class $k$. Then, the vacuity is evaluated as $K/\sum_{k=1}^K(e_{ik}+1)$ as given in Eq. 1.

---

### Official Review · Reviewer_Wp6W · 2022-10-27

**Confidence:** 4
**Correctness:** 3
**Technical Novelty And Significance:** 3
**Empirical Novelty And Significance:** 3
**Recommendation:** 5

**Clarity, Quality, Novelty And Reproducibility:**

Clarity: The paper is clear, however, some detailed descriptions of networks need to be improved.
Quality: The paper is technically sound, and some claims are supported by the experiments.
Novelty: This paper provides a novel idea that integrates reinforcement learning with evidential learning to perform exploration.
Reproducibility: The repository of the source code is expired.


**Strength And Weaknesses:**

Strengths
1) The authors propose a new recommendation framework, i.e., DERL, which integrates a customized RNN and an EAC (evidence-actor-critic) module to exploit dynamic state space and explore item space, respectively.
2) The proposed method leverages vacuity, an second-order uncertainty, to perform effective exploration.
3) The authors have performed extensive experiments on real datasets to demonstrate the effectiveness of the proposed method.

Weaknesses
1) In Section 1, the third contribution claimed by the authors is "An off-policy formulation to effectively promote the reuse of previously collected data while stabilizing model training, which is important to address data scarcity in recommender system". However, in this work, there is no experimental analysis demonstrating the proposed method can address the data scarcity challenge.
2) In Figure 1 and Table 1, the examples show that existing methods fail to capture the user's dynamic preference to obtain the maximum reward. However, the description about the user's long-term preference is missing in these cases.
3) In Section 4.3, for the evidence network, the authors introduce that "e_{ik} is the evidence collected for rating k for item i". However, it is not clear how to collect the learned evidence e_{ik} from the current action a_t and the item pool I using the evidence network.
4) The description about the experimental datasets are not clear enough. For example, the number of interactions in the pre-processed Netflix dataset, the number of users and the number of items on Yahoo dataset.
5) Even though this paper includes many baselines, some classical top-N item recommendation methods should be considered as baselines, e.g., BPRMF and LightGCN. For dynamic recommendation models, the authors need to consider the dynamic top-N recommendation methods as baselines, for example, Dynamic poisson factorization, RecSys 2015.
6) The RL-based recommendation baselines, i.e., DRN, LIRD, and CoLin, are published in 2018 or earlier. The authors need to consider state-of-the-art RL-based recommendation methods as baselines. More RL-based recommendation methods can be found in these surveys: 1) Reinforcement learning based recommender systems: A survey, ACM Computing Surveys 2022, 2) A Survey on Reinforcement Learning for Recommender Systems, arXiv:2109.10665.
7) Does the input feature of the model contain category information (e.g., movie genre)? What factors can help improve the diversity of genre in the evidence-based exploration?
8) The link to the source code is expired.





**Summary Of The Paper:**

The paper proposes a deep evidential reinforcement learning framework, which aims to learn a more effective recommendation policy by integrating both the expected reward and evidence-based uncertainty. Specifically, a recurrent neural network (RNN) is used to exploit dynamic state space, which generates the current state by incorporating past (previous state information), current (sliding window interactions), and future (recommended items from the RL agent) interests. Besides, an EAC module (evidential-action-critic network) is used to perform evidence-based exploration by maximizing a uniquely designed evidential Q-value to derive an optimal policy. Experimental results on real-world data have been performed to demonstrate the effectiveness of the proposed model.

**Summary Of The Review:**

This paper provides a novel reinforcement learning-based recommendation framework. However, the experimental analysis seems not convincing enough. The baselines used in the experiments are not strong enough.

---

> ### Author Response · Authors · 2022-11-18
> **Response to Reviewer Wp6W [Part I]**
>
> Thank you for reviewing our paper and providing valuable comments. We summarize our response as follows.
>
> **Q1. Experimental analysis demonstrating the proposed method can address the data scarcity challenge.**
>
> We would like to clarify that the proposed framework focuses on providing dynamic recommendations. Different from a static setting, there are usually much fewer user interactions in each time step in a dynamic setting. In our experiments on all four real-world datasets, each time step contains no more than 10 interactions for each user, which is highly sparse given a large item space. In order to ensure a stable training of the RL agent and overcome the data scarcity, we develop an off-policy formulation that effectively uses previously collected data through a replay buffer and stabilizes the training of the RL agent. The outstanding recommendation performance clearly justifies that the proposed approach effectively tackles the data scarcity problem in the dynamic setting.
>
> **Q2.  In Figure 1 and Table 1, missing users’ long-term preferences**
>
> Thank you for the suggestion. We refer to user's long-term preference as the genres of movies that the user frequently interacted in the later phase of interactions ($i.e.,$ after time step 8 in the given example). These genres are Musical, Action, Adventure, Comedy, and Animation, which match almost perfectly with what DERL recommends in Table 1.
>
> **Q3. How to collect the learned evidence $e_{ik}$ and item Pool $I$**
>
> To generate evidence $e_{ik}$ of rating class $k$ for interacted item $i$, the evidence network (as shown in Figure 2) takes as input the current action  $a_t$ and embedding of item $i$ and then output the corresponding evidence value. The item pool contains all the interacted item embeddings, where the embeddings are obtained by a pre-trained feature embedding network. Training of the evidence network is given in Section 4.3.
>
> **Q4. Description of the experimental datasets**
>
> We have provided a detailed description of all the datasets in the beginning of Appendix D.
>
> **Q5. Recent standard recommendations methods**
>
> Thank you for suggesting these important baselines. Given the evidential nature of the proposed framework, it predicts the evidence for each rating class for any given item. As a result, it is primarily applicable for recommendation problems with explicit ratings. Because of this, we mainly compare with existing recommendation models that are designed to handle explicit ratings. In this regard, methods, such as BPRMF and LightGCN, are not directly applicable to explicit rating datasets. For LightGCN, we have managed to modify its original implementation to fit explicit rating datasets and provide a comparison with our approach. The results in following table shows worse performance of LightGCN method in both evaluation metrics as compared with our method due to its incapability to handle sparse datasets in a dynamic setting. Similarly, the DPF model is designed specifically for the count-based datasets so it is not applicable to our settings.
>
> | **Model**     | **MovieLens-1M**|**MovieLens-1M** |**MovieLens-100K** |**MovieLens-100K** |
> |  -- | -----------  |--|--|--|
> |     | **P@5** |**nDCG@5** |**P@5** |**nDCG@5** |
> | LightGCN | 0.5668|0.4884|0.5714|0.4852
> |**DERL**|**0.6313$\pm$0.058**|**0.5365$\pm$0.031**|**0.6379$\pm$0.061**|**0.5386$\pm$0.034**|
>
> We would like to further clarify that in addition to the nine comparison baselines that we have presented in the main paper, we also include three extra recent baselines, including REINFORCE, DeepFM and DCNv2 in Table 5 of Appendix G.2.
>
>
>
> **Q6. RL-based recommendation baselines**
>
> Thank you for the suggestion. We have already shown the results on Movielens-1M and Movielens-100K in Table 5 of Appendix G.1 using the REINFORCE model [1] from the survey paper [2] as the reviewer suggested. REINFORCE applies off-policy learning to handle data scarcity. But it has limited exploration power and hence its performance is worse than the proposed DERL method. Here we include the results on DRR [3] and DHCRS [4] as two additional latest RL based recommendation baselines and present the comparison result as in the table below. The obvious advantage of exploration over these baselines further confirms
> its outstanding recommendation performance achieved by DERL.
>
> | **Model**     | **MovieLens-1M**|**MovieLens-1M** |**MovieLens-100K** |**MovieLens-100K** |
> |  -- | -----------  |--|--|--|
> |     | **P@5** |**nDCG@5** |**P@5** |**nDCG@5** |
> | DRR [3] | 0.5370 | 0.5112 | 0.5890 |  0.5225 |
> | DHCRS [4] | 0.4980 | 0.5080 | 0.5214 | 0.4746 |
> |**DERL**|**0.6313$\pm$0.058**|**0.5365$\pm$0.031**|**0.6379$\pm$0.061**|**0.5386$\pm$0.034**|

---

> ### Author Response · Authors · 2022-11-18
> **Response to Reviewer Wp6W [Part II]**
>
> **Q7. Category information in input feature and factors to improve the diversity of genre in exploration**
>
> We have used genre information in the input feature for two Movielens datasets but not used in Netflix and Yahoo datasets as such information is not available.
> To improve diversity, our uniquely design evidential Q-value (see Eq. 4) employs evidence-based uncertainty to explore items with a high predicted rating (the first term in Eq. 4) as well as a high information value (the second term in Eq. 4). In particular, the information value is captured by vacuity, which reflects the lack of knowledge on the item by the RL agent. Intuitively, for a movie with a genre that that RL agent has never seen before, a high vacuity will be generated, which will increase the evidential Q-value. As a result, such a movie is more likely to be selected that improves the diversity of the genre.
>
> **Q8. Expired link for the source code**
>
> Sorry for the inconvenience. We have updated the link, which can be accessed here: https://anonymous.4open.science/r/EvidentialRecommendation-2BDE/README.md.
>
> **References**
>
> - [1] Chen, Minmin, et al. "Top-k off-policy correction for a REINFORCE recommender system." Proceedings of the Twelfth ACM International Conference on Web Search and Data Mining. 2019.
>
> - [2] Afsar, M. Mehdi, Trafford Crump, and Behrouz Far. "Reinforcement learning based recommender systems: A survey." ACM Computing Surveys (CSUR) (2021).
>
> - [3] F. Liu, R. Tang, X, et al. "State representation modeling for deep reinforcement learning based recommendation" Knowl.-Based Syst., vol. 205, p. 106170, October 2020.
>
> - [4] M. Fu et al. Deep Reinforcement Learning Framework for Category Based Item Recommendation. IEEE Transactions on Cybernetics, 2021.

---

### Decision · Program_Chairs · 2023-01-20

**Decision:**

Reject

**Justification For Why Not Higher Score:**

Based on the reviews and the subsequent discussions, I think it's fair to say that this paper is not ready for publication yet given there are still concerns about the methodological contribution as well as the choice of baselines.

**Justification For Why Not Lower Score:**

N/A

**Metareview: Summary, Strengths And Weaknesses:**

Summary: This paper presents a deep evidential reinforcement learning (DERL) approach to sequential recommendation. The basic idea is to add evidential uncertainty around the expected reward to encourage exploration (similar to UCB). On top the standard actor-critic framework with an actor network and a critic network, an evidence network is also incorporated to compute a so-called evidential Q-value and DERL optimizes policy based of it. Experimental results on public benchmark datasets demonstrate the effectiveness of the proposed approach.

Strength: All the reviewers agree that exploration in recommender systems is an important task. The idea of using evidential uncertainty to encourage exploration is interesting. Extensive experiments on real datasets demonstrate the effectiveness of the proposed method.

Weaknesses: The reviewers have some general concerns about the choice of baselines and empirical studies. Specifically, reviewer JmPi argues that more sophisticated exploration strategies should be compared against. Moreover, even though the authors included some additional RL baselines during the discussions phase, none of them have exploration in it. There are also questions regarding the technical contribution of the work given its more applied nature, which I think is a fair point. Finally, the writing can be improved: for example, evidence is huge part of the paper yet 2 out of the 3 reviewers had trouble understanding how exactly the evidence is computed.

One reviewer gave a high score but also acknowledged the issues pointed out by the other reviewers and wouldn't necessarily champion the paper.